# Maternal Supply of Fatty Acids during Late Gestation on Offspring’s Growth, Metabolism, and Carcass Characteristics in Sheep

**DOI:** 10.3390/ani11030719

**Published:** 2021-03-06

**Authors:** Milca Rosa-Velazquez, Jerad R. Jaborek, Juan Manuel Pinos-Rodriguez, Alejandro Enrique Relling

**Affiliations:** 1Facultad de Medicina Veterinaria Zootecnia, Universidad Veracruzana, 91710 Veracruz, Mexico; zS18015915@estudiantes.uv.mx (M.R.-V.); jpinos@uv.mx (J.M.P.-R.); 2Department of Animal Sciences, Ohio Agricultural Research and Development Center (OARDC), The Ohio State University, Wooster 44691, OH, USA; jaborek.1@osu.edu

**Keywords:** fatty acids, fetal programming, insulin sensitivity, prepartum diet, sheep

## Abstract

**Simple Summary:**

Growth is an important factor that drives animal production, and it can be manipulated through maternal nutrition. In ruminants, previous studies suggested that maternal nutrition during late gestation with polyunsaturated fatty acids altered growth, energy metabolism, muscle development, and body composition of the offspring. This study investigates the effect of supplementing different sources of fatty acids during late gestation on offspring energy metabolism and growth during the finishing period. Maternal fatty acid supplementation during late gestation modified growth, insulin sensitivity, and hot carcass weight in lambs; these changes depended on the unsaturation degree of the fatty acid supplement and lamb sex. Hence, fatty acid supplementation of the gestating ewe can potentially have a lifelong impact on offspring’s growth performance and metabolism, and could be used as a possible management alternative to enhanced offspring productivity.

**Abstract:**

Lambs born from dams supplemented with different sources of fatty acids (FA) during late gestation have a different growth rate and plasma glucose concentration. The main objectives of this experiment were to evaluate the effect of supplementing different sources of FA during late gestation on offspring plasma metabolite concentrations, growth, and on a glucose tolerance test (GTT) during the finishing phase. Fifty-four lambs (18 pens, 3 lambs/pen) were born from ewes supplemented during late gestation with one of three treatments: (1) no FA (NF); (2) a source of monounsaturated FA (PDS, 1.01% of Ca salts); or (3) a source of eicosapentaenoic acid (EPA) and docosahexaenoic acid (DHA) (EDS, 1.01% of Ca salts containing). At birth (day 0), supplementation ceased, and all ewes and lambs were placed in a common pen. On day 60, lambs were weaned, grouped by sex, blocked by body weight (BW), and placed on a common finishing diet for 54 days (FP). One lamb per pen was used for the GTT after the FP. There was a tendency for FA × Sex × Day interaction (*p* = 0.08) on lamb growth during the finishing period, with PDS females being heavier than PDS males, while EDS males were heavier than EDS females at day 60. There was a tendency for FA × Sex interaction (*p* = 0.06) for plasma insulin concentration for the GTT. Plasma insulin concentration of wethers increased as FA unsaturation degree increased during the GTT; the opposite happened with the plasma insulin concentration of female lambs. In conclusion, FA supplementation during late gestation tended to modified growth and insulin response to a GTT; these changes differed with the degree of FA unsaturation of the supplement and lamb sex.

## 1. Introduction

Nutrition during gestation is key for adequate intrauterine development, which may significantly impact the offspring’s physiology, health, and metabolism. Studies conducted in humans have demonstrated a correlation between circulating maternal fatty acids (FA) and developmental epigenetic programming [1,2]. Therefore, maternal nutrition during gestation is considered an extrinsic stimulus leading to fetal programming [3,4]. Moreover, studies conducted in animal models have documented the beneficial effects of maternal dietary supplementation of polyunsaturated fatty acids (PUFA) on offspring’s energy metabolism [5], development, growth [6], and inflammatory response [7]. Hence, nutritional management of the gestating dam is extremely important for food animal production because of its potential lifelong impact on health and growth performance; and it could be used as a possible management alternative to enhance offspring productivity.

Previous studies conducted in ruminants have reported the potential effects of PUFA supplementation during late gestation on energy metabolism, growth, body composition [8,9,10], and inflammatory response [8,11] of the offspring. Fatty acid supplementation during late gestation also changed offspring post-weaning growth rate [9,10]. However, these studies did not compare the effect of different FA sources with different degrees of unsaturation to a non-supplemented FA control diet. The treatments fed in these previous studies were a diet enriched with PUFA or a diet enriched saturated fatty acids (SFA) and monounsaturated fatty acids (MUFA) [9], or supplementation with increasing amounts of PUFA [10]. Consequently, we cannot confirm that the difference observed in growth is because of a positive effect of the dietary PUFA, or a negative effect of the supplementation with SFA and MUFA.

An experiment conducted in beef cattle [8], where a PUFA supplement was provided during late gestation, showed a greater body weight (BW) and average daily gain (ADG) of the offspring, which could be due to a greater dry matter intake (DMI); however, DMI was not reported. Previous studies [12] have reported that maternal dietary fat content modulated regulation of the offspring’s appetite. Interestingly, no differences were observed for DMI in studies performed in sheep [9]. Nevertheless, Nickels et al. [10] reported that the offspring’s increased growing BW, from PUFA dam supplementation during late gestation, was positively associated to the lamb’s plasma glucose concentration and not to DMI. Moreover, none of aforementioned experiments were designed to evaluate glucose metabolism. Nickles et al. [10] suggested that a possible impaired activity in glucose, insulin, and ghrelin homeostasis might be responsible for the greater finishing BW of the offspring. Although, it remains unknown which metabolic changes are responsible for the greater growth rate observed in offspring born to PUFA-supplemented dams.

Omega-3 PUFA supplementation of late gestating beef cows has also shown an improvement in carcass quality (greater hot carcass weight, greater longissimus dorsi muscle area, and greater marbling score) of their offspring [8]. These effects were not observed in lambs born to ewes supplemented with PUFA during late gestation [9].

Studies conducted in murine models, where a source of omega-3 PUFA was supplemented, reported greater concentrations of anti-inflammatory-protective lipid mediators. Some of the mediators were resolvins and enzymes involved in their metabolic pathway [13,14] which was associated with an increase in fetal growth [14], and correlated to a reduction in necroinflammatory liver injury [13]. In ruminants, supplementation with a source of omega-3 PUFA to ewes during late gestation showed a greater mRNA expression of arachidonate 5-lipoxygenase activating protein, one of the enzymes involved in the last step of formation of these anti-inflammatory-protective lipid mediators in the fetal liver [11]. Additionally, maternal supplementation with PUFA during late gestation affected the offspring’s plasma haptoglobin concentration, a protein involved in the acute-phase reaction [8]. These results [8,11,13,14] suggest that maternal supplementation with PUFA during late gestation can modulate the offspring’s inflammatory response and result in a greater growth rate of the offspring.

In ruminants, previous studies observed that maternal nutrition during late gestation with PUFA altered growth, metabolism, muscle development, and body composition of the offspring. Therefore, we hypothesized that supplementing omega-3 PUFA during the third trimester of gestation will modulate energy metabolism and inflammatory pathways, which will improve postnatal performance of the offspring. The objective of the current experiment was to evaluate the effects omega-3 PUFA enriched diet to pregnant ewes during late gestation on the offspring’s growth, energy metabolism, plasma resolvin D1 (RvD1) concentration, and carcass characteristics.

## 2. Materials and Methods

### 2.1. Experimental Design and Sampling

The current experiment was conducted at the Sheep Research Center of the Ohio Agricultural Research and Development Center, Wooster, Ohio, under The Ohio State University Institutional Animal Care and Use Committee procedure #2019A00000001.

A scheme of the sampling protocol can be found on Figure 1.

Fifty-four pregnant Dorset × Hampshire ewes (initial BW = 96.2 ± 14.60 kg; 3 to 4 years old) were housed in 18 pens (3 ewes per pen) and supplemented from day 100 of gestation until lambing (day -50 and day 0, respectively). Day one of pregnancy was considered the day on which estrus was confirmed as described previously [10]. Ewes were blocked by BW, and within each block randomly assigned to one of three treatments: (1) Ewes were fed a basal diet with no supplemental lipids to meet sheep nutrient requirements during late gestation [15] (NF); (2) Ewes were fed the same basal diet as NF and supplemented with Ca salts of a palmitic fatty acid distillate (PDS) as a source of palmitic and oleic acids at 1.01% of their dry matter intake; and (3) Ewes were fed the same basal diet as NF and supplemented with Ca salts containing eicosapentaenoic acid (EPA) and docosahexaenoic acid (DHA) (EDS) at 1.01 % of their dry matter intake. The Ca salts were EnerGII and Strata G113 for the PDS and EDS, respectively (Virtus Nutrition LLC, Corcoran, CA, USA). Ewes were fed 2.02 kg/day of diet containing the different treatments (Table 1). The dose of FA supplementation was based on previous fetal programming experiments where supplementation at amounts of FA supplement similar to the ones in this experiment affected offspring growth without affecting dam performance [9,10,16,17]. The sources of FA were chosen because they are rumen inert, commercially available, and contain similar amounts of net energy and Ca. Commercially, there is no rumen inert product that contain purified EPA and DHA. The PDS was chosen to be a commercial product rich in saturated (SFA) and monounsaturated fatty acids (MUFA) (Table 2). As described previously [10], both diets with PDS and EDS have 0.54% more net energy for maintenance (NE_m_) than the NF, but the supplementation with a new feed ingredient to make isoenergetic diet in the current experiment might confound the effects of the supplementation with the different sources of FA vs. effects caused by another feed ingredient, as described previously [18]. Even though the diets were not isoenergetic, an increase of 0.54% in NE_m_ intake per treatment is likely not enough to cause the offspring changes observed in the current experiment. Feed samples were taken weekly and pooled for further analysis. No orts (feed refusals) were registered during the entire research period for the dams. At birth (day 0), the supplementation stopped and all ewes and lambs were placed in a common pen and feed ad libitum the same diet until weaning (day 60). Measurements of ewe BW and body condition score (BCS) were taken at day -50 and day -9. For the BCS the sample was assessed using a 5-point scale [19]. Blood samples were taken from ewes at lambing (day 0).

On day 0, blood samples were collected from lambs between 8 to 16 h post-lambing depending on whether the delivery happened overnight or during the daytime. Thus, lamb blood samples at birth were collected after colostrum consumption. With the occurrence of twins or triplets, one lamb was randomly selected. On day 30 lambs were weighed. On day 60, lambs were weighed, blood sampled, weaned, and blocked by body weight (heavy, medium, and light blocks). Lambs had an adaptation to a common pelleted finishing diet. After the adaptation, the finishing diet was fed for 54 days (Table 3). Starting day of the finishing period was staggered a week between blocks. Therefore, the starting days for the finishing period were day 67, day 74, and day 81 for the heavy, medium, and light blocks, respectively. The initial day of the finishing period, lambs (*n* = 54) were weighed, allotted by sex, and placed in 18 pens (3 lambs/pen; 6 pens/treatment with 3 pens with ewes and 3 pens with wethers). Lamb BW and blood samples were taken on the initial day of the finishing period, and 28 and 54 days after the start of the finishing period (iBW, 28 day BW and fBW, respectively). Blood samples of the light blocks for the last sampling were not analyzed due to problems with the temperature after processing.

Blood samples from dams (10 mL) and lambs (6 mL at birth and 10 mL the rest of the samplings) were taken from the jugular vein, transferred into polypropylene tubes (14 mL, VWR International, Radnor, Pa). The polypropylene tubes contained solutions of disodium EDTA (1.6 mg/mL of blood) and benzamidine HCL (4.7 mg/mL of blood). The tubes were and immediately placed on ice and then centrifugated for 25 min at 1800× *g* at 4 °C. Once the tubes were centrifugated, plasma was aliquoted in individual micro polypropylene tubes with snap caps (1.5 mL, VWR International, Radnor, PA, USA) and stored at −80 °C until analysis. At birth, both dam and offspring blood samples were analyzed for plasma glucose and non-esterified fatty acids (NEFA) concentrations. Offspring blood samples at birth were also analyzed for plasma total FA, and plasma resolvin D1 (RvD1) concentration.

Fifty-five days after the beginning of the finishing period, 18 lambs (one/pen, six/treatment) were selected randomly to conduct a glucose tolerance test (GTT). Catheters (Milacath^®^ extended use #1603, 16 g × 3.0 inches, Mila International, Inc., Florence, KY, USA) were placed in the jugular vein of each lamb using aseptic procedures. Lambs were restrained, their necks were sheared on the right side, the sheared area was rinsed with 70% (vol/vol) ethyl alcohol, scrubbed with surgical soap, and rinsed again with ethyl alcohol. The rinsing and scrubbing procedure was conducted three times. Under local anesthesia (Lidocain 2% -lidocaine hydrochloride injection, MWI/VetOne, Boise, ID, USA), a skin incision of approximately 1 cm was made over the jugular vein with a sterile scalpel. The catheter was placed, and surgical glue and suture were used to stabilize it. A Vetrap bandage was used to protect and keep the catheter in place. Lambs were moved into individual pens without feed (for 24 h) and provided ad libitum access to water.

Prior to conducting the GTT, lambs were weighed 30 min before glucose administration to determine the glucose bolus size (0.25 g of glucose/kg of BW in a 50 % wt/vol dextrose solution). Blood samples were collected 5 min prior to glucose administration (−5 min) and at 2, 5, 10, 15, 20, 30, 60, and 90 min after glucose administration. Before each blood sample was collected and to remove all catheter content blood (1 mL) was collected in a spare syringe and discarded. Blood samples of 10 mL were collected; and subsequently, 1 mL of heparin solution (10 IU of heparin/mL and 0.9% NaCl) was infused into the catheter to prevent clotting. With the 10 mL of blood collected, 7 mL were transferred into a polypropylene tube (VWR International, Radnor, Pa) similar to the ones used for the ewes blood collection (containing disodium EDTA and benzamidine HCL), to measure plasma insulin and ghrelin concentrations, and 3 mL were transferred into 4 mL BD Vacutainer plastic tube with Fluoride (0268847, Fisher Scientific, Pittsburg, PA, USA) to evaluate plasma glucose concentration. Both tubes were immediately placed in ice and centrifuged using the same protocol to separate plasma from blood as explained previously. Plasma from tubes was aliquoted and stored micro polypropylene tubes with snap caps (1.5 mL, VWR International, Radnor, Pa, USA) at −80 °C until analyzed for glucose and insulin. The procedure used to determine plasma ghrelin concentration was similar; however, plasma was placed in similar tubes that were acidified with the addition of 50 μL of 1 N HCl and 10 μL of phenylmethylsulfonyl fluoride per 1 mL of plasma. Plasma glucose and insulin concentrations were measured at −5, 2, 5, 10, 15, 20, 30, 60, and 90 min after the glucose bolus administration; while ghrelin plasma concentration was measured at −5, 2, and 5 min relative to the glucose bolus administration.

Fifty-six days after the beginning of the finishing period, another group of 18 lambs (one/pen was randomly selected, six/treatment) were harvested at the Meat laboratory (Department of Animal Sciences, Ohio State University, Columbus, OH, USA). At harvest, hot carcass weight (HCW) was recorded. Carcasses were stored overnight at 4 °C in a walk-in cooler prior to recording carcass data. Carcasses were ribbed between the 12th and 13th rib after the 12-h chill (4 °C), then a trained university employee determined ribeye area (REA), back fat thickness (BFT), body wall thickness (BWT), and marbling score.

### 2.2. Sample Analysis

A pooled feed sample was analyzed according to the AOAC (1990) for dry matter (DM, method number 981.10), crude protein (CP, method number 967.03), neutral detergent fiber (NDF), and acid detergent fiber (ADF) according to Van Soest et al. (1991) with a heat-stable amylase included in the NDF. Total FA composition of Ca salts was determined using the methods described by Weiss and Wyatt [20].

Plasma total FA were extracted as described by Folch et al. [21] with few modifications [22]. The extracted FA were methylated as described by Doreau et al. [23]. All fatty acid methyl esters were separated by gas chromatography (GC, model HP 5890) using a CP-SIL88 capillary column (film thickness: 100-m × 0.25-mm × 0.2-µm; Varian Inc., Palo Alto, CA, USA). Resolvin D1 concentration in lamb’s plasma collected at birth was quantified with the use of a commercial kit (Resolvin D1 ELISA, Cayman Chemical, Ann Arbor, MI, USA). We conducted the validation for the assay to be use for sheep plasma, based on a parallel displacement. For it we used serial dilutions of ovine plasma compared to the RvD1 standard curve. The recovery for RvD1 was 101.03 ± 3.02%. The intraassay and interassay coefficients of variation were 5 and 14%, respectively. The RvD1 assay was conducted according to the manufacturer’s protocol with slight modifications. Based on the parallel displacement results, we decide not extract the samples. Samples were diluted at a ratio of 1:1 RvD1 assay buffer to plasma to fit the values in the standard curve. Plasma glucose (1070 Glucose Trinder, Stanbio Laboratory, Boerne, TX, USA) and NEFA concentrations were measured using a commercial ELISA immunoassay kit (NEFA Wako Pure Chemical 1, FUJIFILM Wako Diagnostics USA Corporation, Richmond, VA, USA) as validated previously for sheep [24]. Intraassay and interassay coefficients of variation were 3.07% and 1.42% for glucose and 3.25% and 0.25% for NEFA, respectively. Plasma insulin concentration was evaluated using a commercial kit (EMD Millipore Corporation, Billerica, MA, USA). The assay was validated for sheep using a parallel displacement as described for the RvD1. The lowest level of insulin detected by this assay was 1.611 µU/mL when using a 100 µL sample size. The interassay coefficient of variation was 8.79%. Plasma acetylated ghrelin concentration was also measured using a commercial assay (Active Ghrelin radioimmunoassay kit # GHRA-88HK, Linco Research, St. Charles, MO, USA) validated for sheep previously [24]. The intraassay coefficient of variation and the minimum sensitivity were 7.12%, 7.8 pg/mL, respectively.

### 2.3. Statistical Analyses

Data from ewe BW, ADG, BCS, and plasma concentrations of glucose and NEFA were analyzed as a randomized complete block (RCB) design. Data was analyzed using the MIXED procedure of SAS (9.4, SAS Institute, Cary, NC, USA.). The model evaluated the fixed effects of FA supplementation, and the random effect of pen within each block. Pen was the experimental unit. Initial BW and BCS were included as covariates for changes in ewe BW and BCS, respectively.

Lamb plasma RvD1 and FA concentrations data were analyzed as a RCB design using the MIXED procedure (SAS Institute, Cary, NC, USA.). The model evaluated the fixed effects of FA supplementation, and the random effect of pen within each block. Lamb sex was added as a second block criteria for the plasma RVD1 concentration.

Lamb BW, ADG, DMI during the finishing period, and plasma concentration of hormones and metabolites data were analyzed as a RCB design with repeated measures using the MIXED procedure (SAS Institute, Cary, NC, USA.). The model tested the fixed effects of treatment, lamb sex, time, and their interactions, and the random effects of pen or lamb (if only 1 lamb was sampled, i.e., GTT and carcass data) within each block. Pen (or lamb, if only one animal per pen was sampled) was considered the experimental unit. Day was included as a repeated measurement. Different covariance structures were compared (compound symmetry, unstructured, autoregressive, and variance components). The compound symmetry structure was used based on the lowest Akaike information criterion. To determine the denominator degrees of freedom for tests of fixed effects, the Kenward Rogers degrees of freedom approximation was used. For lab data before weaning, type of birth (single or multiple) was included as covariate and removed if it was not significant (*p* > 0.1)

Least square means (LSMEANS) and standard errors of the mean (SEM) were determined using the LSMEANS statement in the MIXED procedure. Significance was set at *p* ≤ 0.05; and tendencies were determined at *p* > 0.05 and *p* ≤ 0.10. The SLICE option of SAS was used for mean separation when the *p*-value for an interaction with time was ≤0.10 and discuss as different if the *p* value of the comparation was ≤0.05. In the case of significant difference on main effect, double or triple interaction, the PDIFF option of SAS was used for mean separation.

## 3. Results

### 3.1. Ewe Performance and Plasma Metabolites

There were no treatment effects (*p* > 0.43) on dam BW, ADG, BCS, and plasma glucose and NEFA concentrations after supplementation with different sources of FA during the last 50 days of gestation (Table 4).

### 3.2. Lamb Performance, Plasma Metabolites, Plasma Fatty Acid Profile, and Plasma Resolvin D1 Concentration at Birth

Dam supplemental treatments, sex of the offspring, and their interactions did not have any effect (*p* ≥ 0.64) on the BW of lambs during the pre-weaning period (Table 5). Despite lamb birth and weaning BW being similar between treatments, a tendency for FA × Day interaction (*p* = 0.09) was observed for lamb plasma glucose concentration. Plasma glucose concentration tended to be greater at birth on the offspring of NF and PDS dams compared with the offspring of EDS dams. However, at weaning, offspring born to EDS dams showed the greatest plasma glucose concentration. There were also tendencies on FA × Day (*p* = 0.08) and Day × Sex (*p* = 0.07) interactions for plasma NEFA concentration. Female lambs born from PDS ewes had the greatest plasma NEFA concentration at birth when compared with the other lambs at either birth or weaning (Table 5).

Regarding plasma FA profile, there was a tendency (*p* ≤ 0.09) for a greater concentration of C17:0, C18:1 t11, and total n-3 in lambs born to EDS ewes (Table 6) compared with lambs born from ewes supplemented with PDS or NF. There was also a tendency (*p* ≤ 0.09) for a lesser plasma concentration of C15:0 ante and C18:1 t6,8 from lambs born from EDS ewes when compared with the lambs born from PDS or NF.

In the present experiment, plasma RvD1 concentration exhibited a FA × Sex interaction (*p* = 0.05; Figure 2). Males born from EDS dams had greater plasma RvD1 concentrations than females born from EDS dams; and females born from PDS ewes showed a greater plasma RvD1 concentration when compared with males born to PDS ewes (Figure 2). Moreover, males born from EDS dams had the greatest RvD1 concentration when compared with the other lambs born from PDS or NF ewes, males born from NF dams showed the lowest RvD1 concentration in comparison with the lambs in the other two treatments.

### 3.3. Lamb Performance and Plasma Metabolites during the Finishing Period

There were tendencies (*p* = 0.08) for FA supplementation and FA × Sex interaction effect for lamb BW during the finishing period; wethers born from EDS ewes were heavier than the lambs in the other two treatments during the whole finishing period after weaning (Table 7). Ewe lambs born from PDS ewes were heavier than wethers born from the same treatment ewes. Lambs born from NF dams had similar BW between sexes. Lambs from NF dams had similar BW than ewe lambs from PDS ewes (Table 7). There was no FA or sex effect (*p* ≥ 0.25) on the lamb’s ADG and plasma metabolites during the entire finishing period (Table 7). No differences were observed for feed intake (*p* ≥ 0.60) or the gain to feed ratio (*p* ≥ 0.16; Table 7).

Carcass characteristics of lambs tended to have a FA × Sex interaction (*p* = 0.07) for HCW; wethers born from EDS ewes had the heaviest HCW when compared to the rest of the lambs in the other two treatments (Table 8). Female lambs born from PDS dams showed a heavier HCW than wethers from PDS ewes but have similar HCW than lambs born from NF ewes (Table 8). However, lambs born from NF ewes were heavier than the wethers born from PDS ewes, while female lambs born from NF ewes were heavier than female lambs born from EDS ewes. Sex of the lamb affected (*p* ≤ 0.01) REA, with the wethers having greater REA compared with female lambs (Table 8). The other carcass characteristics (dressing percentage, BFT, BWT, and marbling) were not affected (*p* ≥ 0.12) by dam FA supplementation or lamb sex (Table 8).

### 3.4. Glucose Tolerance Test

Plasma glucose concentration of lambs during the GTT (time effect, *p* ≤ 0.01) was greatest 2 min after the glucose bolus administration. Plasma glucose concentration decreased thereafter, until 90 min after bolus administration (Figure 3). There were no FA × Sex or FA × Time interactions (*p* ≥ 0.14) observed for plasma glucose concentration during the GTT. On the other hand, a three-way interaction was observed for insulin concentration. There was a tendency for FA × Time × Sex interaction (*p* = 0.08; Figure 4) for plasma insulin concentration. Wethers’ plasma insulin concentration tended to increase as FA unsaturation degree increased during the GTT, while the plasma insulin concentration decreased with increasing FA unsaturation for female lambs. Ewe lambs born from PDS dams tended to have a greater insulin concentration than wethers born from PDS ewes (Figure 4). Conversely, wethers born from PDS ewes showed their greatest plasma insulin concentration at 5 min after bolus administration (Figure 4). Plasma insulin concentration during the GTT for ewe lambs born from PDS ewes showed their greatest concentration at 90 min after bolus administration (Figure 4B). Wethers born from NF dams showed their greatest plasma insulin concentration at 10 min after bolus administration. Ewe lambs born from NF dams showed their greatest insulin concentration at 10 min after bolus administration (Figure 4B). Wethers born from EDS dams showed a greater increase in insulin concentration at 15 min after bolus administration, when compared with the other two treatments (Figure 4B), whereas ewe lambs born from EDS dams showed their greater insulin concentration at 5 min after the bolus administration (Figure 4B). No dam FA supplementation, lamb sex, time, two- or three-way interactions (*p* ≥ 0.20) were observed for lamb plasma ghrelin concentration during the GTT (Figure 5).

## 4. Discussion

### 4.1. Ewe Performance and Plasma Metabolites

Our results are in accordance with those reported by Nickles et al. [10] and Coleman et al. [16], who fed up to 2% of Ca salts of FA containing omega-3 PUFA during the last 50 days of gestation and did not report any differences in ewe performance or plasma metabolites. However, when different sources of FA are supplemented during the peripartal period, the results are inconsistent. Sheibani et al. [25] did not observe any differences in ewe BW or plasma metabolite concentrations when a Ca soap of fish oil (FO) was supplemented from four weeks until three weeks after parturition. Nonetheless, Capper et al. [26] reported a decrease in ewe BW at lambing when ewes were fed a diet supplemented with 6% Ca salts of palm oil instead of FO from day 103 of gestation into early lactation. Thus, the difference between our results and the ones reported in these two aforementioned studies could be due to the source, amount, and period of the FA supplementation.

### 4.2. Lamb Performance, Plasma Metabolites, Plasma Fatty Acid Profile, and Plasma Resolvin D1 Concentration at Birth

Maternal FA supplementation did not change lamb birth or weaning BW. Other studies performed with ruminants reported contradictory results when different sources of FA were supplemented during late gestation. No changes in BW during the pre-weaning period were reported in lambs born to ewes supplemented with a source of SFA or PUFA during late gestation [16], or a source of SFA or FO 6 weeks pre-partum until 4 weeks post-partum [26]. However, Garcia et al. [27] observed a greater BW at birth in calves born to cows fed a diet supplemented with either SFA or Ca salts containing PUFA during the last 8 weeks of pregnancy when compared with calves born to cows that consumed a diet with no FA supplementation. In the experiment by Garcia et al. [27], calves born from SFA-supplemented dams were heavier and had a greater DMI than the calves born from PUFA-supplemented cows, with maternal FA supplementation occurring during the peripartal period. The difference in our results with the ones aforementioned [26,27] could be due to the period of FA supplementation, as well as, the FA source, profile, and amount of FA supplemented, and the differences between the species used in these experiments.

The tendency for a greater plasma glucose and NEFA concentrations observed in the present study at birth in lambs and a greater birth and weaning plasma NEFA concentration in female lambs born to PDS-supplemented dams disagrees with what has been reported in previous studies [10,16]. The management and growth rate from birth to weaning in the current experiment and in the previous ones [10,16] are similar. However, the current experiment compared the effect of sex and its interaction, which none of the other research does. Despite Nickles et al. [10] reporting there was no sex effect for the growing variables, the addition of sex and its interaction on the model can remove some of the error, which allowed us to observe the effect of treatment over time for plasma glucose and NEFA concentrations. Currently, we do not have a physiological explanation of why the trend on changes in plasma metabolite concentrations were not associated with changes in body weight for the different treatments.

On the other hand, it has been reported that the plasma FA profile of the lamb can be affected by maternal dietary FA supplementation during gestation [22,27], and can also reflect the FA that are consumed in colostrum [22,28]. Thus, we expected that lambs born to dams fed an omega-3 PUFA enriched diet during late gestation would have a greater long-chain PUFA concentration in blood plasma at birth. A greater supply of PUFA can be used by the newborn to improve its development and health status, since they are involved in a wide range of biological processes including cellular proliferation and differentiation [29], proper immune function [7,30], and enhanced fetal development [14]. From the long chain SFA reported in lamb plasma at birth, two of them were affected by maternal FA supplementation. Lambs born to EDS ewes tended to have a lesser C15:0-ante concentration and a greater C17:0 plasma concentration. It has been reported that changes in plasma FA profile are reflective of the FA composition of dietary supplements consumed by the individual [26]. However, in the present study, differences in the lamb’s plasma FA do not reflect the FA profiles of the Ca salts consumed by the dam during late gestation. Thus, the greater concentration of C15:0-ante in the plasma from lambs born to PDS ewes, and the greater concentration of C17:0 observed in lambs born from EDS ewes, can be an indicator of an increased production of these FA in the dam’s rumen due to MUFA and PUFA supplementation. These FA could have been later transferred to the offspring via placenta during late gestation or after colostrum consumption. On the other hand, two of the C18:1 isomers were affected by maternal FA supplementation. Maternal supplementation with PUFA tended to result in a lesser C18:1 t6,8 concentration and in a greater C18:1 t11 concentration. It has been reported by Bauman and Griinari [31] that EPA and DHA can inhibit rumen biohydrogenation, which results in the formation of trans isomers. Therefore, the differences observed with these two C18:1 isomers in the offspring’s plasma could be due to changes in dam FA biohydrogenation caused by dietary omega-3 PUFA supplementation, which changed these trans C18:1 modifying their concentration in maternal plasma, and possibly transferred into the offspring at the end of gestation or via colostrum. Despite the lack of effect of dam treatment on lamb EPA and DHA plasma concentration at birth, a numerically greater concentration for these two omega-3 FA in lambs born from EDS ewes, which could have contributed to the greater concentration of total n-3 FA, was observed. Other studies reported inconsistent results regarding EPA and DHA plasma concentrations in lambs born to ewes supplemented with a source of EPA and DHA during gestation [22,28,32]. Supplementation with a low dose (0.39% of the DMI) of EPA and DHA during late gestation did not influence the concentration of EPA and DHA in lamb plasma at birth [22]. It is important to note that sampling was taken after the first milking; therefore, the consumption of colostrum might have confounded some of the results of the differences on plasma FA concentration due to placenta passage [22]. However, other studies, where a greater amount of EPA and DHA was supplemented during gestation, reported a greater concentration of EPA and DHA in the plasma of lambs at birth [28,32] (4.73% and 4.5%, respectively]. Therefore, a greater supplementation rate of EPA and DHA can be required to elicit an effect on the FA profile of lamb plasma at the time of birth. Moreover, and due to the aforementioned important functions of these FA in the animal biological processes [29,30,33], it is also a possibility that the dams could have held on to these essential FA instead of transferring them through the placenta or via colostrum to the offspring.

Omega-3 EPA (20:5n-3) and DHA (22:6n-3) may influence the inflammatory response [34]. These omega-3 PUFA can be substrate for potent anti-inflammatory and pro-resolving lipid mediators named resolvins [34,35]. Previous studies suggest that maternal exposure to dietary omega-3 PUFA could increase these lipid mediators, which may have beneficial effects in the inflammatory response and subsequent offspring development. Dietary supplementation with omega-3 PUFA increased resolvin’s basal concentrations in the rodent liver [13], placenta [14], bone marrow [36], and in human blood [37] when compared to subjects that did not receive dietary omega-3 PUFA; however, resolvin basal concentration after omega-3 PUFA supplementation has not been studied in ruminants prior to this study. Moreover, previous studies [14,36] focus on maternal resolvin concentration in different tissues after dietary supplementation during gestation. Thus, to our knowledge, the present study is the first to evaluate the effect of both the offspring’s sex and dam’s FA supplementation on offspring plasma RvD1 concentration. Our data suggest that offspring plasma RvD1 concentration at birth can be affected by source of dietary FA supplemented during gestation and the sex of the newborn. The changes observed in RvD1 concentration at birth in the present study were positively associated with changes in offspring BW during the finishing period. Hence, the possible modulatory effect of maternal PUFA supplementation during late gestation on a newborn’s inflammatory response could have had a positive effect in the lamb’s growth during finishing period.

### 4.3. Lamb Performance and Plasma Metabolites during the Finishing Period

Wethers born from dams supplemented with a source of EPA and DHA trend to have the greatest post-weaning BW during the finishing period, which also resulted in a tendency for greater HCW. Even though the observed differences in offspring BW decreased over time during the finishing period, they were positively associated with the tendencies observed in HCW. Carranza-Martin et al. [9] and Marques et al. [8] reported heavier offspring weight in animals born to dams supplemented with a diet enriched with Ca salts of EPA and DHA when compared with offspring born to dams supplemented with Ca salts of SFA or MUFA. Marques et al. [8] also observed a greater HCW on steers born from PUFA-supplemented cows. However, in the study conducted by Marques et al. [8] only male offspring were used, thus no sex effect was taken into consideration. In the study conducted by Carranza-Martin et al. [9], no sex effect was reported, but they did not evaluate the interaction between offspring sex and FA supplementation. Our results suggest that FA supplementation affects offspring performance in a sex dependent manner. Ewe lambs have a greater BW when their dam consumes a gestational diet supplemented with MUFA, while wethers have a greater BW when their dam consumes a diet enriched with PUFA during late gestation. The data in the present study also suggest that maternal supplementation with MUFA during late gestation can have a limiting effect on male offspring development, and similar for female offspring whose dam is supplemented with a PUFA gestational diet. The observed differences in lamb BW were not associated with differences in average daily gain, plasma metabolites concentration, feed intake, or feed conversion (gain to feed ratio) during the post-weaning period, and neither of these variables were affected by the dam’s supplemental treatment. A previous study, where increasing amounts of PUFA (0 vs. 1 and 2%) were fed during late gestation [10], demonstrated a greater lamb BW and greater plasma glucose concentration were positively associated with increasing PUFA supplementation to dams during late gestation. On the other hand, offspring plasma glucose concentration differences were not reported by Carranza-Martin et al. [9] where a lesser amount of PUFA supplementation was fed during gestation (0.39% of DM) when compared with a supplementation with SFA. Nickles et al. [10] also reported a positive correlation between plasma glucose concentration in lambs born to PUFA-supplemented dams. However, the changes on plasma glucose concentration reported previously [10] were not associated with a greater DMI. Based on the results from Nickles et al. [10], and the ones reported in a previous study [18], Nickles et al. [10] assumed that maternal PUFA supplementation modulated offspring glucose metabolism; and that the changes observed in the offspring finishing weight can be due to an unpaired glucose-insulin system, suggesting a need for future research in this area. Nickles et al. [10] also found that offspring ghrelin plasma concentration of PUFA born lambs was negatively associated with offspring plasma insulin concentration; and it could probably be causing the observed changes in lamb finishing BW. Something worth to mention is that in the current experiment energy intake of the dams or the lambs from lambing to weaning was not measured. It might be possible that FA supplementation during gestation might affect DMI and energy intake in the dam, which might affect milk yield and lamb growth. However, previous studies [10,16] showed no difference in milk yield on milk nutrient composition in similar sheep models.

### 4.4. Glucose Tolerance Test

Based on the results of the two aforementioned studies [10,18], a GTT was conducted during the finishing period where plasma concentration of glucose, insulin, and ghrelin was measured. Different from what was reported by Nickles et al. [10], dam FA supplementation did not have any effect on plasma glucose or ghrelin concentration during the GTT. Furthermore, no association was established between plasma glucose and insulin concentration, and no FA by sex interaction was observed for ghrelin and glucose plasma concentrations. Our results indicate that dam FA supplementation during late gestation affects the glucose-insulin system in a sex dependent manner’ suggesting that males born from PUFA-supplemented ewes are less sensitive to insulin than females born to ewes fed the same FA during gestation. This lack in insulin response was similar in female lambs born from MUFA-supplemented ewes. Furthermore, these changes in the glucose-insulin system were associated with the changes observed in lamb BW during the finishing period.

## 5. Conclusions

Maternal FA supplementation during the last trimester of gestation affected offspring plasma RvD1 at birth and trend to affect growth and energy metabolism, all in a sex-dependent manner. Wethers born from PUFA-supplemented ewes had a better performance (greater BW and HCW) and were less insulin sensitive, while ewe lambs born from MUFA-supplemented ewes had better development (greater BW and HCW) and were less insulin-sensitive than males born to MUFA-supplemented ewes. This tendency in differences in lamb BW and plasma insulin concentration was not associated with lamb DMI but was positively associated with RvD1 plasma concentration at birth. Our results may suggest that wethers born to PUFA-supplemented dams during late gestation will be able to deal better with inflammatory processes and will trend to have a greater growth, the same happening with ewe lambs born from MUFA-supplemented ewes.

## Figures and Tables

**Figure 1 animals-11-00719-f001:**
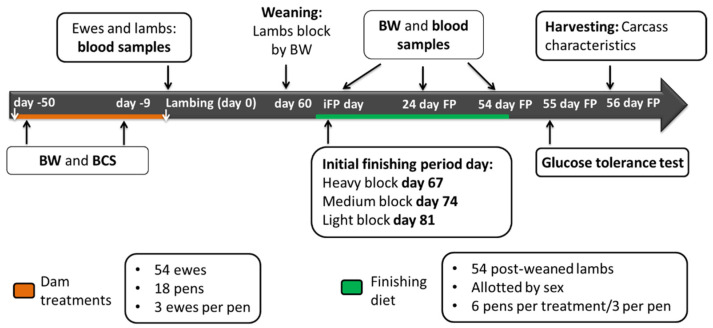
Timeline scheme of the feeding and sampling conducted to the dams (ewes) and the offspring.

**Figure 2 animals-11-00719-f002:**
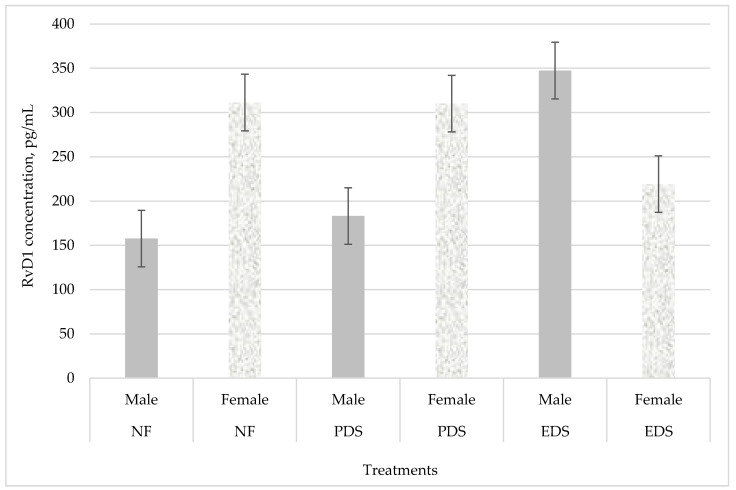
Effect of a fatty acid enriched diet during the last 50 days of gestation and offspring’s sex on plasma RvD1 concentrations at birth. Treatments of ewes supplemented with Ca salts of MUFA (PDS; EnerGII, Virtus Nutrition LLC, Corcoran, CA, USA.), PUFA (EDS; Strata G113, Virtus Nutrition LLC, Corcoran, CA, USA.) or no supplementation (NF) during the last 50 days of gestation (54 lambs, 3 dam treatments). Data are presented as the LSmean of the FA × Sex (*p* = 0.05; standard error of the mean = 80.37). No difference (*p* ≥ 0.31) for treatments or sex.

**Figure 3 animals-11-00719-f003:**
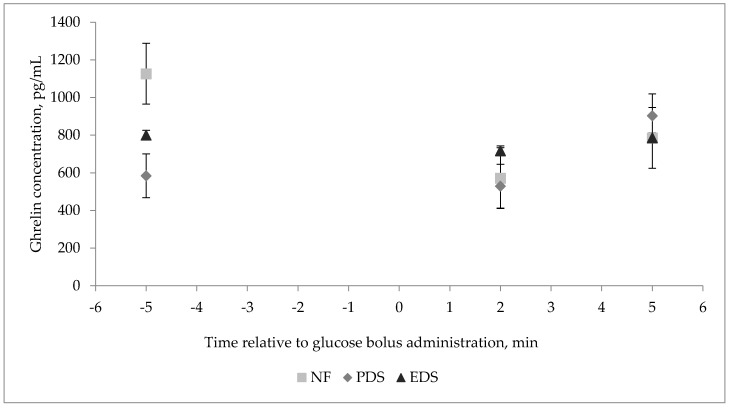
Effect of a fatty acid enriched diet during the last 50 days of gestation and offspring’s sex on glucose concentration during the glucose tolerance test on offspring during the finishing period (18 lambs; 2 sex and 3 dam treatments). Data are presented as the LSmean of FA × Time (*p* = 0.34; standard error of the mean (SEM) = 13.64); time (*p* ≤ 0.01; SEM = 8.16). No difference (*p* ≥ 0.14) for FA, sex, and FA × Sex. Treatments of ewes supplemented with Ca salts of MUFA (PDS; EnerGII, Virtus Nutrition LLC, Corcoran, CA), PUFA (EDS; Strata G113, Virtus Nutrition LLC, Corcoran, CA), or no supplementation (NF) during the last 50 days of gestation.

**Figure 4 animals-11-00719-f004:**
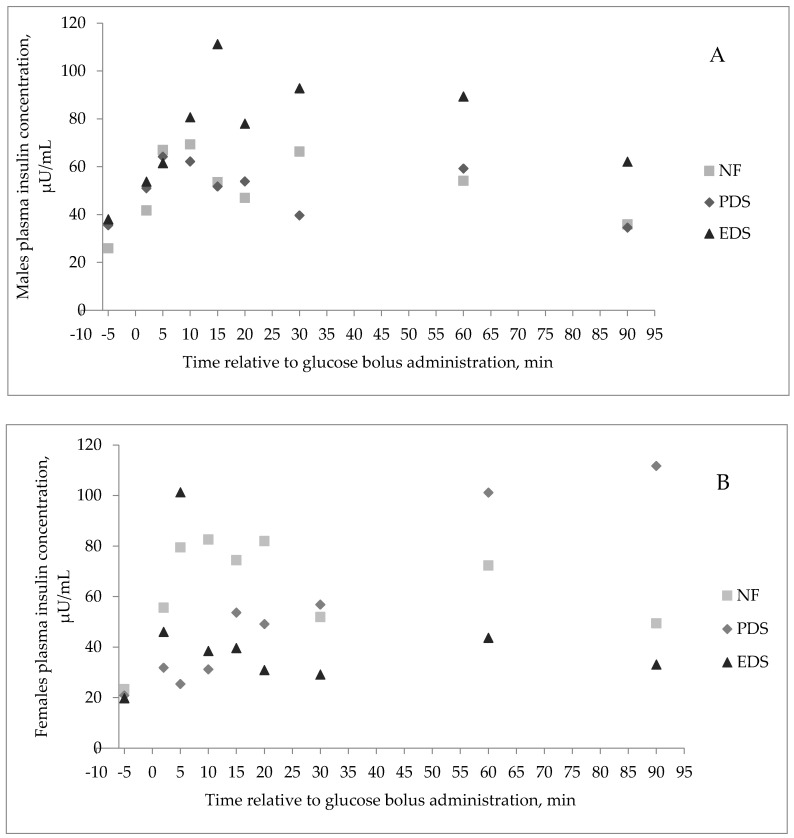
Effect of a fatty acid enriched diet during the last 50 days of gestation and offspring’s sex on plasma insulin concentration during the glucose tolerance test on offspring during the finishing period (18 lambs; 2 sex and 3 dam treatments; (**A**) is the response for male and the (**B**) is the response for females). Data are presented as the LSmean of Fatty acid supplementation × Time × Sex (*p* = 0.08; standard error of the mean (SEM) = 20.03). No difference (*p* ≥ 0.47) for FA, Time, Sex, FA × Time, and Time × Sex. Treatments of ewes supplemented with Ca salts of MUFA (PDS; EnerGII, Virtus Nutrition LLC, Corcoran, CA), PUFA (EDS; Strata G113, Virtus Nutrition LLC, Corcoran, CA), or no supplementation (NF) during the last 50 days of gestation.

**Figure 5 animals-11-00719-f005:**
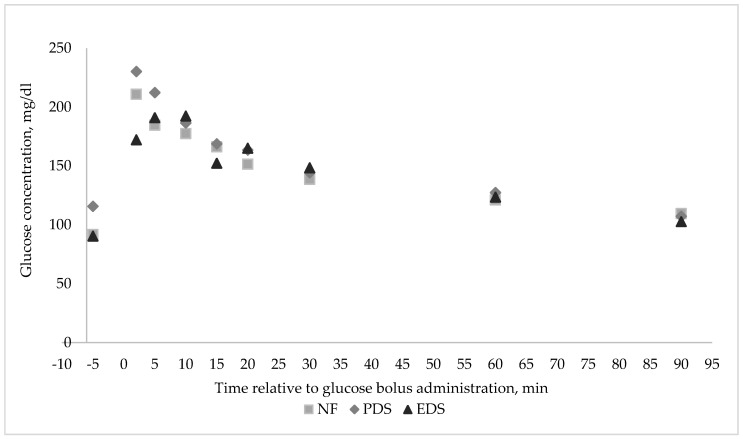
Effect of a fatty acid enriched diet during the last 50 days of gestation on plasma ghrelin concentrations during the glucose tolerance test on offspring during the finishing period (18 lambs; 2 sex and 3 dam treatments). Treatments of ewes supplemented with Ca salts of MUFA (PDS; EnerGII, Virtus Nutrition LLC, Corcoran, CA), PUFA (EDS; Strata G113, Virtus Nutrition LLC, Corcoran, CA), or no supplementation (NF) during the last 50 days of gestation. Data are presented as the LSmean of fatty acids × Time (*p* = 0.29; standard error of the mean = 205.43). No difference (*p* ≥ 0.20) for FA, Time, Sex, Time × Sex, FA × Sex, and FA × Time × Sex.

**Table 1 animals-11-00719-t001:** Diet ingredients and chemical composition (% DM basis) of the dietary treatments fed to pregnant ewes at 2.02 kg/day during the last 50 days of gestation.

	Treatment (% of Dry Matter Basis)		
	NF ^1^	PDS ^2^	EDS ^3^
Ingredient			
Alfalfa haylage	17.96	17.96	17.96
Corn silage	30.54	30.54	30.54
Ground Corn	10.10	10.10	8.89
DDGS ^4^	10.10	10.10	10.07
Limestone	0.44	0.44	0.48
Soy Hulls	30.65	30.65	30.88
PUFA Ca Salts	-	-	1.01
MUFA Ca Salts	-	1.01	-
Mineral and Vitamin Pre-mix ^5^	0.20	0.20	0.18
Chemical Composition			
Neutral detergent fiber	40.70	44.08	41.79
Crude protein	15.46	15.50	17.99
Ether extract	3.68	5.28	4.37
Ash	5.87	6.09	6.40

^1^ NF ewes daily rations meet nutrient requirements (NRC, 2007). ^2^ EnerGII, Virtus Nutrition LLC, Corcoran, CA. ^3^ StrataG113, Virtus Nutrition LLC, Corcoran, CA. ^4^ Dried distiller’s grains with solubles. ^5^ Vitaferm Concept-Aid Sheep (BioZyme, St. Joseph, MO). Contains Ca (15.5%), P (5%), NaCl (16%), Mg (4% Mg), K (2%), Co (10 ppm), I (70 ppm), Mn (2850 ppm), Se (16.4 ppm), Zn (2500 ppm), Vitamin A (130,000 IU/kg), Vitamin D3 (7500 IU/kg), and Vitamin E (550 IU/kg).

**Table 2 animals-11-00719-t002:** Fatty acid profile (% of total fatty acid) of Ca salts used to supplement pregnant ewes during the last 50 days of gestation.

Fatty Acid	Supplement ^1^	
	PDS	EDS
C8:0 + C10:0 + C12:0	0.62	0.12
C14:0	1.17	5.99
C16:0	45.87	22.01
C16:1	0.20	7.40
C18:0	5.14	7.47
C18:1 c9	36.27	17.46
C18:1 other	1.10	4.51
C18:2	8.03	2.69
C20:0	0.37	0.34
C20:1	0.09	0.84
C18:3	0.20	0.94
C22:0	-	0.35
C22:1	-	1.38
C20:3 n-3	-	0.51
C20:5	0.13	9.19
C22:6	-	7.00
Other	0.80	12.15

^1^ PDS: source of palmitic and oleic acid; EDS: source of eicosapentaenoic acid and docosahexaenoic acids (Ener GII and Strata G113, respectively; Virtus Nutrition LLC, Corcoran, CA). Fatty acid profiles evaluated using the methods of Weiss and Wyatt.

**Table 3 animals-11-00719-t003:** Diet formulation and nutrient composition (expressed as % of DM basis) fed to the offspring of the pregnant ewes supplemented during late gestation (last 50 days) with no additional fatty acid or 1.01% of ca salts containing MUFA (PDS ^1^) or PUFA (EDS ^1^).

	% of Dry Matter Basis
Ingredients	
Corn (dry rolled)	62.59
Soyhulls	24.07
Soybean meal	11.08
Mineral and Vitamin Pre-mix ^2^	2.36
Chemical Composition	
Neutral detergent fiber	16.24
Crude Protein	15.60
Ether extract	3.45
Ash	4.78

^1^ PDS: source of palmitic and oleic acid; EDS: source of eicosapentaenoic acid and docosahexaenoic acids (Ener GII and Strata G113, respectively; Virtus Nutrition LLC, Corcoran, CA, USA). ^2^ Mineral and vitamin mix contains (as % DM basis): 19.35 urea, 38.76 limestone, 19.36 sodium chloride, 0.4 Vitamin A, 0.4 Vitamin D3, 1.96 Vitamin E, 3.76 selenium, 0.49 Bovatec 91, and 15.52 Ammonium chloride.

**Table 4 animals-11-00719-t004:** Effect of a fatty acid enriched diet during the last 50 days of gestation on ewes’ body weight (BW), body condition score (BCS), average daily gain (ADG), and plasma metabolites.

Item	Treatment ^1^			SEM ^2^	*p*-Value
	NF	PDS	EDS		FA
Ewes (Pens)	18 (6)	18 (6)	18 (6)		
BW, kg ^3^	97.81	97.44	96.72	2.33	0.78
ADG, kg/day ^3^	0.25	0.24	0.22	0.06	0.78
BCS ^4^	2.93	3.16	3.05	0.12	0.43
Glucose, mg/dL	90.77	80.40	92.49	10.37	0.68
NEFA, µEq/L	316.40	283.15	385.58	76.28	0.64

^1^ Treatments of ewes supplemented with Ca salts of SFA and MUFA (PDS; EnerGII, Virtus Nutrition LLC, Corcoran, CA), EPA and DHA (EDS; Strata G113, Virtus Nutrition LLC, Corcoran, CA, USA), or no supplementation (NF) during the last 50 days of gestation. ^2^ SEM = standard error of the mean. ^3^ Initial body weight was used as a covariate. ^4^ Initial body condition score was used as a covariate; body condition score measured on a scale of 1–5.

**Table 5 animals-11-00719-t005:** Effect of a fatty acid enriched diet during the last 50 days of gestation on body weight (BW), and plasma metabolites from birth to weaning in male (M) and female (F) lambs.

Treatment ^1^	NF		PDS		EDS		SEM	*p*-Values				
Lamb Sex ^2^	F	M	F	M	F	M		FA	Sex	FA × Sex	FA × Day	Day × Sex
Item												
Lambs (pens)	9 (3)	9 (3)	9 (3)	9 (3)	9 (3)	9 (3)						
BW, kg ^3^							0.76	0.75	0.74	0.96	0.64	0.95
Birth (day 0)	5.5	5.5	5.4	5.9	5.3	5.4						
day 30	15.9	16.8	16.7	16.2	16.8	17.2						
day 60	26.9	26.5	26.5	26.4	27.2	27.7						
Glucose, mg/dL ^4^							15.74	0.33	0.49	0.70	0.09	0.43
Birth (day 0)	129.10 ^a^	154.12 ^a^	125.05 ^a^	108.72 ^b^	115.75 ^b^	104.38 ^c^						
day 60	69.06 ^c^	77.35 ^b^	58.46 ^c^	67.23 ^c^	81.63 ^b^	109.67 ^a^						
NEFA, µEq/L ^4^							122.40	0.07	<0.01	0.16	0.08	0.07
Birth (day 0)	988.41 ^b^	852.84 ^b^	1248.64 ^a^	645.73 ^c^	696.96 ^c^	523.62 ^c^						
day 60	398.91 ^a^	365.96 ^b^	431.44 ^a^	297.01 ^c^	366.66 ^b^	359.12 ^b^						

^1^ Treatments of ewes supplemented with Ca salts of SFA and MUFA (PDS; EnerGII, Virtus Nutrition LLC, Corcoran, CA, USA.), EPA and DHA (EDS; Strata G113, Virtus Nutrition LLC, Corcoran, CA, USA.), or no supplementation (NF) during the last 50 days of gestation; SEM= standard error of the mean. ^2^ Females (F), and males (M). ^3^ There was no FA × Day × Sex interaction (*p* = 0.77). ^4^ There was no FA × Day × Sex interaction (*p* ≥ 0.49). a,b,c. Values with different superscripts differ with a *p* ≤ 0.05.

**Table 6 animals-11-00719-t006:** Effect of a fatty acid enriched diet during the last 50 days of gestation on lambs’ plasma fatty acid concentration (% total fatty acid methyl esters) at birth.

Item	Treatments ^1^			SEM ^2^	*p*-Value
	NF	PDS	EDS		FA
Lambs	18	18	18		
C4:0	0.18	0.04	0.06	0.06	0.30
C6:0	0.26	0.23	0.17	0.11	0.80
C10:0	0.34	0.52	0.30	0.08	0.25
C12:0	0.78	0.96	0.57	0.14	0.17
C14:0	5.30	6.75	4.32	1.01	0.25
C14:0 iso	-	-	0.02	0.02	0.44
C15:0 iso	0.17	0.12	0.21	0.07	0.66
C15:0 ante	0.27	0.33	0.14	0.06	0.09
C15:0	0.75	0.73	0.71	0.07	0.92
C16:0 iso	0.15	0.09	0.20	0.06	0.41
C16:0	28.20	30.49	26.30	1.70	0.23
C17iso	1.37	1.31	1.01	0.18	0.32
C16:1 & C17:0 ante	3.37	3.68	3.22	0.35	0.63
C17:0	1.16	0.94	1.25	0.09	0.09
C17:1	0.70	0.58	0.59	0.05	0.24
C18:0	10.89	9.40	10.59	0.77	0.35
C18:1 t6,8	0.85	0.71	0.56	0.09	0.07
C18:1 t9	0.15	0.21	0.27	0.06	0.31
C18:1 t10	1.31	1.32	2.76	0.61	0.13
C18:1 t11	0.62	0.51	1.69	0.38	0.06
C18:1 t12	0.66	0.74	0.55	0.13	0.59
C18:1 c9	31.62	29.21	26.59	3.39	0.50
C18:1 c11	2.26	1.64	2.30	0.40	0.43
C18:1 c12	0.27	0.20	0.12	0.08	0.38
C18:1 c15	0.04	0.07	-	0.03	0.25
C18:1 c16	0.27	0.30	0.28	0.09	0.97
C18:2 c9,c12	4.41	4.88	9.97	3.36	0.37
C18:3	0.15	0.25	0.22	0.08	0.65
C20:0	0.10	0.25	0.19	0.05	0.21
C20:1	0.20	0.28	0.45	0.13	0.42
C18:2 9c,11t	0.47	0.56	0.73	0.16	0.43
CLA2 other	0.11	0.09	-	0.07	0.43
C22:0	0.18	0.07	0.18	0.06	0.28
C20:3 n-6	0.05	0.12	0.21	0.06	0.20
C22:1	1.62	1.54	1.47	0.29	0.94
C20:5 n-3	0.23	0.17	0.56	0.16	0.18
C24:0	0.12	0.08	0.21	0.07	0.36
C22:6 n-3	0.49	0.55	1.01	0.18	0.11
Identified peaks	99.99	99.99	99.99	0.0004	0.96
Total MUFA	40.15	37.45	34.70	3.77	0.52
Total PUFA	5.33	5.96	11.92	3.67	0.31
Total n-3 ^3^	0.86	0.97	1.74	0.31	0.08
Total n-6 ^3^	4.47	4.99	10.18	3.41	0.36
Total EPA and DHA ^3^	0.71	0.72	1.5	0.32	0.11
n-6/n-3	6.67	5.94	4.38	1.91	0.64

^1^ Treatments of ewes supplemented with Ca salts of MUFA (PDS; EnerGII, Virtus Nutrition LLC, Corcoran, CA), PUFA (EDS; Strata G113, Virtus Nutrition LLC, Corcoran, CA), or no supplementation (NF) during the last 50 days of gestation. ^2^ SEM= standard error of the mean. ^3^ CLA = conjugated linoleic acid; n-3 = omega 3; n-6 = omega 6; EPA = eicosapentaenoic acid; DHA = docosahexaenoic acid.

**Table 7 animals-11-00719-t007:** Effect of a fatty acid enriched diet during the last 50 days of gestation on offspring development and plasma metabolites during the finishing period in male (M) and female (F) lambs

Treatment ^1^	NF		PDS		EDS		SEM	*p*-Values		
Lamb Sex	F	M	F	M	F	M		FA	Sex	FA × Sex
Item										
Lambs (pens)	9 (3)	9 (3)	9 (3)	9 (3)	9 (3)	9 (3)				
BW, kg ^2^							4.08	0.08	0.72	0.08
Initial	32.84 ^b^	32.06 ^b^	31.45 ^b^	29.21 ^c^	31.93 ^b^	34.45 ^a^				
28 day BW	41.02 ^b^	40.45 ^b^	39.59 ^b^	37.70 ^c^	39.24 ^b^	42.86 ^a^				
FBW	48.56 ^b^	48.33 ^b^	47.02 ^b^	46.09 ^c^	46.82 ^b^	50.30 ^a^				
ADG, kg/day ^3^							0.06	0.68	0.25	0.86
FP1	0.29	0.30	0.29	0.30	0.26	0.30				
FP2	0.29	0.29	0.28	0.32	0.29	0.28				
DMI, kg ^3^							0.21	0.76	0.86	0.60
FP1	2.73	2.70	2.75	2.68	2.58	2.88				
FP2	3.25	3.02	3.25	3.31	2.98	3.08				
G:F ^3^							0.02	0.68	0.16	0.51
FP1	0.24	0.24	0.23	0.25	0.22	0.23				
FP2	0.19	0.21	0.19	0.21	0.21	0.20				
Glucose, mg/dL ^4^	68.75	61.71	71.53	71.89	69.95	66.94	9.41	0.78	0.67	0.92
NEFA, µEq/L ^4^	102.34	134.13	149.20	138.70	156.16	99.11	34.48	0.76	0.70	0.49

^1^ Treatments of ewes supplemented with Ca salts of MUFA (PDS; EnerGII, Virtus Nutrition LLC, Corcoran, CA, USA.), PUFA (EDS; Strata G113, Virtus Nutrition LLC, Corcoran, CA, USA.), or no supplementation (NF) during the last 50 days of gestation; SEM= standard error of the mean. ^2^ Body weight measure at the starting of the finishing period (initial), 28 days into the finishing period (28dBW), and 54 days from the start of the finishing period (FBW). There were no FA × Day, Day × Sex, or FA × Day × Sex interactions (*p* ≥ 0.35). ^3^ FP1 consists of the sampling between the initial body weight and day 28 of the finishing period. FP2 consists of the sampling between day 28 and day 54 sampling of the finishing period. There were no FA × Day, Day × Sex, or FA × Day × Sex interactions (*p* ≥ 0.60). ^4^ There were no Day × Sex or FA × Day × Sex interactions (*p* ≥ 0.60). a,b,c. Values with different superscripts differ with a *p* ≤ 0.05.

**Table 8 animals-11-00719-t008:** Effect of a fatty acid enriched diet during the last 50 days of gestation on male (M) and female (F) lamb carcass characteristics.

Treatments ^1^	NF		PDS		EDS		SEM	*p*-Values		
Lamb Sex	F	M	F	M	F	M		FA	Sex	FA × Sex
Item										
Hot carcass weight, kg	27.81 ^b^	28.26 ^b^	27.81 ^b^	26.07 ^c^	26.14 ^c^	30.30 ^a^	3.45	0.49	0.33	0.07
Dressing, %	0.53	0.54	0.56	0.54	0.54	0.56	0.02	0.67	0.86	0.43
Back fat thickness, cm	0.74	0.53	0.56	0.56	0.74	0.61	0.04	0.37	0.12	0.39
Body wall thickness, cm	2.44	2.08	2.29	2.08	2.39	2.18	0.12	0.92	0.18	0.95
Ribeye area, cm^2^	15.55 ^b^	17.10 ^a^	14.71 ^c^	16.19 ^b^	13.29 ^c^	17.16 ^a^	0.17	0.47	<0.01	0.35
Marbling score ^2^	10.00	10.33	9.67	9.67	9.67	10.67	0.54	0.36	0.19	0.44

^1^ Treatments of ewes supplemented with Ca salts of MUFA (PDS; EnerGII, Virtus Nutrition LLC, Corcoran, CA, USA.), PUFA (EDS; Strata G113, Virtus Nutrition LLC, Corcoran, CA, USA.), or no supplementation (NF) during the last 50 days of gestation (18 lambs, 2 sex, 3 dam treatments); SEM = standard error of the mean. ^2^ Marbling score is based on a scale: 9 = slight, 10 = small. ^a,b,c.^ Values with different superscripts differ with a *p* ≤ 0.05.

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
