# Peer review of "Maternal Supply of Fatty Acids during Late Gestation on Offspring’s Growth, Metabolism, and Carcass Characteristics in Sheep"

_animals, 2021, doi:10.3390/ani11030719_

Round 1

Reviewer 1 Report

I believe this paper is suitable for Animals journal, and I suggest to accept it after some revision. My concerns were listed as below: 1) Please draw a figure to present the animal experiments and sample time point, which could helpful to understand the experimental procedure. 2) in all tables, you marked a,b,c Values with different superscripts differ with a P ≤ 0.05 for main effects and P ≤ 0.10 for interactions. Why you select P ≤ 0.10 for interactions, I believe P ≤ 0.05 for interactions should be better?

Author Response

Please see the response to reviewers in the attached file.

Reviewer 2 Report

This article by Rosa-Velazquez et al focusing on prenatal maternal nutrition and its effect on offspring. It is an ambitious project, and includes 54 lambs, and multiple endpoints. Specifically, its aim is to identify the effect of the supplementation of maternal ewe's diet with polyunsaturated fatty acids on offspring weight and other physiologic measures. The comparator groups included a non-supplemented diet and a diet supplemented with monounsaturated fatty acids. The authors reach many conclusions, including the assertion that fatty acid supplementation during late gestation modified growth and insulin tolerance.   There are numerous concerns with this work, both major and minor.   Major concerns: 1) The statistical analysis for this paper asserts interactions should be considered significant at a p-value of <0.1. This opens up the work to a significant risk of type one error. This is especially true given that drawing conclusions from interaction terms opens up the work to significant risk of the multiple comparison problem, and any definitive conclusions drawn from these interactions should probably be subjected to a multiple comparison statistical correction. Acceptance of this work should be predicated on peer review by a statistician that comments on these issues.    2) Sample sizes for each time point and measurement are not clear in figures and tables. Standard error of the mean is presented, which is reliant on sample size. It is more appropriate to present standard deviations. Also, when figures are presented, it would be helpful if each individual was presented as a data point with a measure of central tendency and deviation overlaid on top.   3) There is the suggestion that offspring sex seriously impacts the results. If this is true, it should be explored, and data should be presented consistently in a sex dependent manner.    Minor concerns: 1. What was the intake of calories after lambs were born?   2. Is there any proposed biologic mechanism for the observed sex differences?

Author Response

Please find response to reviewer in the attached file. 

Round 2

Reviewer 2 Report

The edited manuscript has been returned. Some of the issues identified in my first review were address, others were not. P-values >0.05 are still suggesting non-significant findings are significant (including multiple times in the abstract and in the conclusion). The suggestion of using SD instead of SEM was not employed (this article may explain to the authors why SD is a more appropriate choice : https://pubmed.ncbi.nlm.nih.gov/23125963/).

I do feel that the lack of information on caloric intake of the lambs should be noted as a limitation

Author Response

The revisions to reviewer 2 are on the attached manuscript
